# Alterations in Healthy Adult Canine Faecal Microbiome and Selected Metabolites as a Result of Feeding a Commercial Complete Synbiotic Diet with *Enterococcus faecium* NCIMB 10415

**DOI:** 10.3390/ani13010144

**Published:** 2022-12-30

**Authors:** Stinna Nybroe, Pernille B. Horsman, Kamilla Krag, Therese G. Hosbjerg, Kathrine Stenberg, Bekzod Khakimov, Jørgen Baymler, Charlotte R. Bjørnvad, Ida N. Kieler

**Affiliations:** 1Department of Veterinary Clinical Sciences, Faculty of Health and Medical Sciences, University of Copenhagen, 1165 København, Denmark; 2Odder Dyreklinik, Banegårdsgade 24, 8300 Odder, Denmark; 3Østfyns Dyrehospital, Skovgyden 15, 5800 Nyborg, Denmark; 4Bacterfield GmbH, 20354 Hamburg, Germany; 5Department of Food Science, Faculty of Science, University of Copenhagen, 1165 København, Denmark

**Keywords:** probiotics, *E. faecium*, dog, faecal quality, cobalamin, cholesterol, short chain fatty acids, microbial richness, microbial relative abundance

## Abstract

**Simple Summary:**

Probiotics are defined as live microorganisms that, when administered in adequate amounts, confer a health benefit on the host. The effect of a commercial diet containing the probiotic *Enterococcus faecium* NCIMB 10415 was examined in a crossover study with 11 healthy privately owned dogs. The dogs were fed the same balanced commercial diet, with or without the probiotic, for 16 days, and then fed the alternate diet for an additional 16 days with a 19 days washout period in between. Owners evaluated their dog’s faecal quality daily. Faecal bacterial composition (microbiome) and short-chain fatty acid concentrations as well as serum concentrations of cholesterol, triglycerides, cobalamin and folate were analysed before and after each feeding period. Soft stools were less frequent when fed the diet with probiotic included compared with the diet alone. The probiotic diet also decreased serum cholesterol. Most observed effects were related to the diet itself, irrespective of probiotic inclusion or not. These effects included an increased faecal microbial diversity and content of the short-chain fatty acids butyrate and acetate, as well as a decrease in serum cobalamin concentration. There were indications of a likely prolonged survival (19 days) of the probiotic strain in a few individual dogs, which has not previously been reported for this probiotic strain.

**Abstract:**

In dogs, the use of probiotics for preventive or therapeutic purposes has become increasingly common, however the evidence for beneficial effects are often limited. The aim of this study was to investigate the effects of feeding a diet containing *Enterococcus faecium* NCIMB 10415 on faecal quality, faecal short-chain fatty acid concentrations, serum concentrations of cholesterol, triglycerides, cobalamin and folate as well as faecal microbiome in adult dogs. Eleven healthy client owned dogs were enrolled in a randomized, double-blinded crossover study. All dogs were fed the same balanced diet with or without incorporation of *Enterococcus faecium* NCIMB 10415 for 16 days each. Blood and faecal samples were collected at baseline and during the feeding trial and owners recorded daily faecal scores. An *Enterococcus* spp. ASV, likely representing *E. faecium* NCIMB 10415 was detected in the faecal microbiome of some dogs 18–19 days after withdrawal of oral supplementation. Inclusion of *E. faecium* decreased circulating cholesterol (*p* = 0.008) compared to baseline. There were no differences in cholesterol concentrations between diets. Owners reported 0.6 ± 0.3) days less of loose stools compared to the control diet. Comparing to baseline, both diets significantly increased faecal concentration of acetate and butyrate, decreased serum cobalamin and increased faecal microbial diversity. Decreased serum cobalamin, and increased faecal acetate correlated with decreases in the *Fusobacterium*, *Streptococcus*, *Blautia*, and *Peptoclostridium*. Except for effects on circulating cholesterol and faecal score, effects were observed regardless of the addition of *E. faecium*. It is therefore likely that these effects can be contributed to dietary prebiotic effects on the faecal microbiome.

## 1. Introduction

In dogs, the use of probiotics for preventive or therapeutic purposes has become increasingly common [1]. Probiotics are defined as live microorganisms that, when administered in adequate amounts, confer a health benefit on the host [2] and a variety of bacterial strains are marketed as probiotics for dogs. Probiotic effects are highly species and strain specific [3,4] and the evidence for their health effect is often limited [4].

Most probiotics alter the gastrointestinal (GI) microbiome composition, favouring lactic acid producing bacteria and creating a more hostile environment for potentially pathogenic Gram-negative bacteria. This is reflected by increased abundance of Lactobacillacea in the faecal microbiome [5,6,7,8,9] and altered faecal SCFA content [5,7,8,9] as a result of supplementing dogs with various strains of probiotics.

Short chain fatty acids support gastrointestinal health as a source of energy and regulator of the microenvironment. Furthermore, studies in dogs indicate that SCFA’s contribute to a probiotic-mediated regulation of circulating cholesterol [10,11,12].

In a recent study, circulating concentrations of cobalamin were below reference range in 44% of dogs 2 weeks after they had been supplemented with a synbiotic containing *E. faecium* NCIMB 10415, trade name SF68 (5 × 10^8^ CFU/g; Fortiflora^®^; Proplan Purina) for two weeks [13]. This is concerning, because dogs rely completely on the dietary content and microbial production of the B vitamins: cobalamin and folate [14]. Since cobalamin is essential for several metabolic processes and has been closely linked with gastrointestinal health and disease, this finding could indicate an adverse effect of probiotic use in healthy dogs.

*E. faecium* NCIMB 10415 (trade name 4b1707) is one of few probiotic strains approved for use in dogs by the European Commission [15]. Currently, a single study, has investigated the effect of *E. faecium* NCIMB 10415, trade name 4b1707 in dogs. In this study, the probiotic was supplemented as a synbiotic with fructooligosaccharide and acacia (gum arabic), and there was a mildly decreased occurrence of stress related diarrhoea in shelter dogs [16]. Although other studies have reported similar improvements in faecal quality with probiotic supplementations [9,17,18], the overall evidence for a positive clinical effect on the faecal quality is still conflicting [4,19]. 

The authors are aware, that until there is sufficient evidence to support the registration of bacterial strains included in diet formulations as probiotics, the European Food Safety Authority (EFSA) apply the term “gut flora stabilizer” instead of “probiotic” for feed additives [20]. However, as the term “gut flora stabilizer” may not be familiar to all, the term “probiotic” will be used for the dietary inclusion of *E. faecium* NCIMB 10415, trade name 4b1707 in the current study.

The current study aimed to investigate the impact of feeding a commercial balanced canine diet with or without *E. faecium* NCIMB 10415, trade name 4b1707 to privately owned dogs for 16 days, specifically evaluating the effects on faecal quality, faecal microbial diversity and relative abundance, faecal SCFA concentrations as well as serum triglycerides, cholesterol, cobalamin, and folate concentrations.

## 2. Materials and Methods

### 2.1. Study Design

The study was designed as a randomized prospective double-blinded crossover study. Prior to initiation, the Danish Animal Experiments Inspectorate (approval number 2018-15-0201-01522), as well as the local administrative and ethical committee at the Department of Veterinary Clinical Sciences, University of Copenhagen, Denmark (approval number 2018-15), ethically approved the study protocol. 

The study consisted of two study periods of 16 days each, with a washout period of 19 days before each study period. Throughout the study, dogs were fed a commercial nutritionally balanced diet. The test diet (+PD) was nutritionally identical to the control diet (−PD), differing only by being manufactured to include *E. faecium* NCIMB 10415 10^9^ CFU/kg. In each study period, the dogs were exclusively fed the allocated diet and in the preceding washout period they were fed the same diet as allocated for the study period. Figure 1 shows an overview of the study outline. 

### 2.2. Animals

Privately owned healthy dogs meeting a set of fixed inclusion criteria were recruited. To be included in the study, the dogs had to be 2–8 years old, have a body weight of at least 10 kg and a body condition score (BCS) of 4/9–6/9 [21]. The dogs had to be clinically healthy based on a physical examination, complete blood cell count (CBC) and biochemistry profile. Chronically ill dogs and dogs receiving medical treatment or having a history of hyporexia or frequent diarrhoea were excluded. Throughout the study, the dogs were housed in their home environment and was cared for by their owners.

### 2.3. Diets

A canine commercially available dry kibble diet “Probiotic LIVE adult with chicken” (Bacterfield GmBH, Hamburg, Germany) was fed to the dogs during the probiotic period (+PD). The +PD diet was manufactured to contain 10^9^ CFU/kg of *E. faecium* NCIMB 10415, trade name 4b1707. The diet was based on hydrolysed chicken protein, rice, maize, chicken fat and salmon oil. Dietary fibre consisting of dried beet pulp (3.7% of dry matter (DM)), dried chicory (1.4% DM) and fructooligosaccharides (1.3% DM) as well as brewer’s yeast and mineral supplements were also added. The probiotic was incorporated into the kibbles after extrusion and drying, by a patented technic, in which the probiotic bacteria are suspended in an oil mixture and driven into the kibble pores by altering ambient pressures in a hermetically sealed chamber. The dietary content of *E. faecium* NCIMB 10415 was controlled and verified by ALS Laboratories (Mirfield, UK) prior to the study. The control diet (−PD) was provided by the same manufacturer and formulated with the same nutritional composition as the +PD, but without the addition of *E. faecium*. The nutritional compositions of the diets as stated by the manufacturer are provided in Table 1.

Daily allowance was initially calculated as Maintenance Energy Requirements (MER [kcal] = 95*BW [kg]^0.75^) for inactive dogs according to the recommendation by the National Research Council [22]. The dogs were weighed weekly and the daily allowance was adjusted when needed to maintain a stable bodyweight throughout the study. Each dog was randomly allocated to a group being fed either −PD or +PD for the first study period. All dogs were gradually transitioned from their current diet to −PD or +PD (whichever diet they were allocated to for the first study period) over a period of 7 days. 

Owners fed their dog according to the calculated daily feeding allowance and dogs had free access to water throughout the study. During the study, no treats, chewing bones or any other feed substances apart from the study diets were allowed. Owners kept formal diaries of their dog’s dietary intake and activity level. 

### 2.4. Faecal Quality

Owners registered consistency for all stools passed during the study periods by using a faecal scoring chart with score 1 being very firm and score 7 being watery diarrhoea [23]. For each study period, faecal quality was evaluated by the number of days on which the owner reported loose stools, defined as faecal score ≥ 4/7 at any point of time throughout the day.

### 2.5. Faecal SCFA and Blood Analysis

At inclusion and after each study period, a physical examination was performed and baseline blood and rectal faecal samples were collected. Blood was collected by venipuncture (cephalic or jugular vein) using a 23G single-use butterfly needle and vacutainer closed system. A trained veterinary technician performed the blood sampling, with minimal restraining of the dog and in the owner’s presence to reassure the dog. The dogs were fasted (minimum 12 h) for all blood collections.

A standard biochemistry profile (Advia 1800 Chemistry System, Siemens, Ballerup, Denmark) and CBC (Adivia 2120i, Hematology System, Siemens, Ballerup, Denmark) was performed at the Veterinary Diagnostic Laboratory, University of Copenhagen, Denmark. Serum was separated and stored at −80 °C within 30 min of collection and serum cholesterol, triglyceride cobalamin and folate concentrations were analysed at the end of the study period (Idexx, BioResearch Ludwigsburg, Germany).

The faecal samples were stored at −80 °C within 30 min of rectal collection and faecal SCFA concentrations were analysed as described previously [24] by gas chromatography mass spectrometry (Agilent Technologies 7890A GC System, Wilmington, DE, USA) at the Department of Food Science, University of Copenhagen, Denmark.

### 2.6. Faecal Microbiom Analysis

In addition to rectal faecal samples for SCFA analysis, the owners collected spontaneous voided faeces on day 0, 24, 25, 40, 41, 59, 60, 75 and 76 for microbiome analysis. These samples were frozen at −20 °C immediately after collection and transferred to −80 °C within 17 days for storage until analysis.

DNA extractions were carried out in a dedicated pre-PCR laboratory and PCR was performed at the Danish National High-Throughput Sequencing Center, University of Copenhagen, Denmark, for details see Appendix B.

Raw reads were quality controlled with FastQC/v0.11.8 [25], demultiplexing, adapter removal, as well as low-quality base removal (minquality  =  30) using AdapterRemoval/v2.2.4 [26]. Proper read-orientation was ensured using cutadapt/v2.6. DADA2 package was then used to trim the reads (error rate algorithm), clustering of the amplicon sequence variants (ASV) (clustering algorithm), and assignment of taxonomy using Silva/v138 (classification algorithm) was performed. Post clustering the LULU algorithm [27] was used to minimize false positive and Decontam package [28] to remove contaminations. Further analysis was then carried out in R version 4.2.0 (R Core Team, 2022), for further details see Appendix C.

### 2.7. Statistics

The software system R version 4.2.0 (2022-04-22) (R Core Team, 2012) was also used to analyse blood and faecal SCFA samples. A separate linear mixed model (lme4 and lmerTest package) was applied with serum cobalamin, folate, cholesterol, triglyceride, acetate, propionate and butyrate concentrations as the outcome, individual dog as the random effect, while diet (−PD vs. +PD) and time period were fixed effects in each model.,. Statistical diagnostics were performed for verification regarding distribution assumptions, and detection of observations with undue influence for each model. A generalized mixed modelling using a Poisson distribution (lme4 package) was used to analyse faecal scores, where diet and time period were included as fixed effects and the individual dog as a random effect. For the faecal scores, two models were used. In the first one score 6–7 were grouped together (diarrhoea), score 2–4 were grouped together (not diarrhoea), while score 1 (too hard) was not modified. In the second model score 4–7 were grouped together (too watery), score 2–3 were grouped together (well formed), while score 1 (too hard) was not modified.

For linear and generalized mixed models *p* < 0.05 was considered significant.

## 3. Results

### 3.1. Dogs

Twenty-six dogs were initially recruited. Nine dogs could not be included due to identified CBC and/or biochemistry abnormalities, which was eosinophilia (n = 5), hypermagnesemia (n = 3) and elevated alkaline phosphatase (n = 1). Additionally, 6 dogs were excluded during the course of the study due to owner request to withdraw from the study (n = 3), diarrhoea with a duration >2 days (n = 2) and weight loss despite increasing daily rations (n = 1). For further details on included/excluded dogs, see Appendix A.

Eleven dogs completed the full crossover study and were included in the final statistical analysis. They were of mixed breed (n = 2), Border Collie (n = 2), Broholmer (n = 1), Field Trial Cocker Spaniel (n = 1), Labrador Retriever (n = 1), Riesenschnauzer (n = 1), Berger Blanc Suisse (n = 1), Greyhound (n = 1) and West Highland White Terrier (n = 1). The population was composed of four entire males, four neutered males and three entire females. Age range was 3.0–7.3 years (median 4.4 years), body weight 10.9–46.3 kg (median 28.8 kg) and body condition score 4–6/9. All dogs were found to be healthy throughout the study based on clinical examination, CBC, and biochemical profile. Prior to entering the study, the dogs were fed a variety of home cooked and commercially available diets (details available in Appendix A).

### 3.2. Faecal Quality

Overall, diarrhoea (defined as faecal score ≥ 6/7) was infrequent, with a total occurrence of 6 out of a total of 338 days for which a faecal score was available and there was no difference between study periods (*p* = 0.42). During the +PD period, the number of days where loose stools (defined as faecal score ≥ 4/7) were reported were 2.5/16 days (range 0/16–10/16), while it was 3.3/16 days (range 0/16–14/16) in the −PD period and statistically the dogs had 0.6 ±0.3 less days with loose stools during the +PD feeding period compared to −PD (*p* = 0.03).

### 3.3. Faecal SCFA

Faecal acetate and butyrate concentrations were significantly increased after feeding either −PD (*p* = 0.002 and *p* = 0.02, respectively) or +PD (*p* < 0.001 and *p* = 0.01, respectively, Figure 2), compared to baseline, with no difference between +PD and −PD. Faecal propionate concentrations were not affected by either diet compared to baseline (Figure 2). 

### 3.4. Serum Lipids

Serum cholesterol concentrations were lower following +PD feeding compared to baseline (*p* = 0.008, Figure 3). There was no difference between serum cholesterol concentrations following +PD and −PD periods or between −PD and baseline. There were no differences in serum triglyceride concentrations between baseline, +PD or −PD (Figure 3).

### 3.5. B Vitamins

Serum cobalamin concentrations were within reference ranges throughout the study, but was significantly lower after feeding both +PD (*p* = 0.002) and −PD (*p* = 0.01) compared to baseline, with no intergroup difference (Figure 4). Serum folate concentrations did not differ between baseline, +PD or −PD feeding (Figure 4).

### 3.6. Microbiome

Observed faecal microbial richness was increased for +PD and −PD when compared to baseline (estimate 241.5, *p* = 0.001 and estimate 216.8, *p* = 0.005, respectively, Figure 5), with no intergroup difference and no effect of feeding +PD prior or after −PD. The Chao1 and Shannon index findings were similar to the observed faecal microbial richness. 

For the beta diversity, no separation of faecal microbial communities was detected relating to the diet or diet order using Bray–Curtis (R^2^ = 0.014, *p* = 0.3 and R^2^ = 0.04, *p* = 0.1, respectively), unweighted UniFrac (R^2^ = 0.02, *p* = 0.2 and R^2^ = 0.03, *p* = 0.1, respectively) or weighted UniFrac (R^2^ = 0.002, *p* = 0.4 and R^2^ = 0.002, *p* = 0.4, respectively). For details on beta diversity, please refer to Appendix A.

Several differences in the proportions of the relative abundance of various ASVs were observed when comparing the effect of feeding +PD to baseline, −PD to baseline, and +PD to −PD. For +PD compared to baseline, *Enterococcus* (ASV313) increased in relative abundance, while *Enterococcus* (ASV335) decreased. The relative abundance of *Enterococcus* (ASV313) also increased with +PD feeding compared to −PD feeding. A relative decrease in the abundance of *Lachnoclostridium* (ASV32) and of *Prevotella* 9 (ASV82) was associated with the +PD and −PD compared to baseline. While *Blautia* (ASV41) only increased with +PD feeding compared to baseline, several *Bacteroides* (ASV63, ASV102, ASV149, and ASV153) increased with −PD feeding compared to baseline. Three of these *Bacteroides* (ASV63, ASV149, and ASV153) decreased with +PD compared to −PD feeding, while three *Blautia* (ASV24, ASV52, and ASV101) increased. In addition, *Bacteroides* (ASV75), *Lachnospira* (ASV16), *Lachnoclostridium* (ASV116), and an unknown genus of Eggerthellaceae decreased with +PD compared to −PD feeding (for further details see Table 2). 

Three ASVs (ASV313, ASV335, and ASV796) belonging to the *Enterococcus* genus were detected throughout the study (Table 3). ASV313 was detected in 18/42 faecal samples from the +PD period regardless of order of the diets but was not present in any baseline samples. For dogs that were fed −PD after +PD, *Enterococcus* ASV313 was present in 3/22 samples from 3 different dogs at the end of the washout period (18–19 days after +PD was no longer provided), but was not present at the end of the −PD period. ASV335 and ASV796 were detected in a few baseline samples as well as a few, +PD and −PD samples when +PD was fed prior to −PD

### 3.7. Correlation between Faecal Microbial Relative Abundance, Fecal Short Chain Fatty Acids and Serum Biomarker Concentrations 

Several ASVs positively correlated with the abundance of faecal SCFA concentrations (Figure 6). For acetate a positive correlation with *Romboutsia* (ASV13, r = 0.65) and *Sutterella* (ASV109, r = 0.6) was found. For propionate, positive correlations with *Sellimonas* (ASV37, r = 0.61), *Allobaculum* (ASV39, r = 0.62) and unknown genus of Lachnospiraceae (ASV68, r = 0.62) were observed. For butyrate, a positive correlation with *Prevotella*_9 (ASV30, r = 0.66) was observed. While for the sum of all SCFAs a positive correlation was observed with *Romboutsia* (ASV13, r = 0.68).

Several negative correlations were observed between the SCFAs and the relative abundance of ASVs. Acetate concentration correlated negatively with *Fusobacterium* (ASV64, r = −0.64), *Streptococcus* (ASV161, r = −0.65), *Peptoclostridium* (ASV217, r = −0.69 and ASV295 r = −0.75), *Blautia* (ASV221 r = −0.70) and three unknown genera of the Peptostreptococcaceae (ASV222, r = −0.73, ASV307, r = −0.77, and ASV363, r = −0.72). For propionate a negative correlation was observed with *Fusobacterium* (ASV64, r = −0.72), three ASVs of the *Peptoclostridium* (ASV104, r = −0.60, ASV165, r = −0.60, and ASV217, r = −0.69), *Prevotella*_9 (ASV112, r = −0.62) and *Streptococcus* (ASV161, r = −0.63). While for butyrate, a negative correlation was observed with *Megamonas* (ASV8, r = −0.74), *Lachnoclostridium* (ASV113, r = −0.6), two *Peptoclostridium* (ASV132, r = −0.60 and ASV295, r = −0.79), *Catenibacterium* (ASV167, r = −0.60), *Blautia* (ASV221, r = −0.76), three ASVs of unknown genera of the Peptostreptococcaceae (ASV222, r = −0.76, ASV307, r = −0.82 and ASV363, r = −0.81) and *Holdemanella* (ASV224, r = −0.64). No negative correlation was observed with the total sum of SCFAs. 

When analysing for possible correlations between serum biomarkers and faecal microbial relative abundance, it should be noted that the faecal microbial ASVs correlating with changes in serum cobalamin and cholesterol levels, reflected by the decrease in serum cobalamin concentration during feeding of the −PD and +PD and the decrease in serum cholesterol concentration during +PD feeding (Figure 7). Positive correlations (r ≥ 0.6) between cobalamin and ASVs belonging to the following genera was found: *Fusobacterium* (ASV64, r = 0.63), *Streptococcus* (ASV161, r = 0.66), two ASVs belonging to the *Peptoclostridium* (ASV217, r = 0.70 and ASV295, r = 0.76), *Blautia* (ASV221 r = 0.71), two of unknown genera of the Peptostreptococcaceae family (ASV222, r = 0.74 and ASV307, r = 0.78), and *Peptostreptococcus* (ASV363, r = 0.73). Positive correlations (r ≥ 0.6) between serum cholesterol and one *Streptococcus* (ASV161, r = 0.61), three unknown genera of the Pepstreptococcaceae (ASV363, r = 0.75, ASV307, r = 0.79, and ASV222, r = 0.74), two of the *Peptoclostridium* genus (ASV217, r = 0.64 and ASV295, r = 0.78), and one *Blautia* (ASV221, r = 0.72). A negative correlation was observed between *Romboutsia* (ASV13) and serum cobalamin (r = −0.67) and cholesterol (r = −0.63), and between *Collinsella* (ASV53, r = −0.6) and serum cobalamin. 

## 4. Discussion

This study investigated faecal SCFAs and microbiome changes as well as selected circulating metabolic markers in relation to feeding healthy client-owned adult dogs a nutritionally balanced dry diet with or without a probiotic incorporated at the level of manufacturing. In addition, a possible interaction between faecal SCFA and microbial changes with circulating concentrations of cobalamin, folic acid, cholesterol, and triglycerides was investigated. Incorporation of *E. faecium* NCIMB 10415 had a cholesterol lowering effect compared with baseline concentrations and slightly improved faecal quality compared to when the dogs were fed the same diet without addition of *E. faecium*. The diets, both with and without *E. faecium* NCIMB 10415, increased the faecal microbiome diversity, changed the specific bacterial relative abundances, and increased faecal butyrate and acetate content. Furthermore, serum cobalamin concentrations decreased with the feeding of the study diets irrespective of *E. faecium* NCIMB 10415 incorporation but remained within reference ranges.

Overall, the diets seemed well tolerated with few reported days of diarrhoea (faecal score ≥ 6/7) for dogs which completed the study. The owner-perceived faecal quality was slightly, but significantly, improved when feeding the diet containing *E. faecium* NCIMB 1041510^9^ cfu/kg compared to without, however, the clinical relevance seems limited. During the study, two dogs were withdrawn due to diarrhoea >2 days. The dogs were privately owned, and suboptimal owner compliance made reintroduction of the diets impossible. Therefore, it is unknown whether it was the diet or other unrelated causes that resulted in diarrhoea in these two dogs. 

The small positive effect on faecal quality is in line with a previous study where the proportion of days with diarrhoea was slightly, but significantly lower in dogs entering a kennel supplemented with a symbiotic containing *E. faecium* NCIMB 10415 (2%) compared with the placebo group (3.2%). 

In the current study, faecal microbial analysis revealed increases in *Enterococcus* ASV313 to be associated with +PD feeding compared to baseline as well as compared to −PD feeding. Two other ASVs belonging to the *Enterococcus* genus (ASV335/ASV796) were also found. However, ASV335 decreased with +PD feeding and none of the diets were significantly associated with ASV796. *Enterococcus* ASV313 is therefore most likely to be *E. faecium* NCIMB 10415 (trade name 4b1707). However, this was not confirmed by for example qPCR with primers specific for the *E. faecium* NCIMB 10415. In a subset of samples from dogs that were allocated to be fed +PD in the first period, ASV313 was detected in the faecal microbiome at the end of the washout period 18–19 days after +PD had been discontinued. This might represent a prolonged survival of *E. faecium* NCIMB 10415 in these individual dogs. However, none of the dogs that had faecal presence of ASV313 by the end of the washout between +PD and −PD had ASV313 present at the end of the −PD study period and it is unknown for exactly how long into the −PD period the ASV313 persisted. 

A prolonged survival of *E. faecium* NCIMB 10415 incorporated into the +PD was not suspected during the planning of the current study as survival beyond the supplementation period, has not previously been described for any *E. faecium* NCIMB 10415 probiotic in dogs (or other species). Based on culturing of rifampin resistant bacteria, prolonged survival has however been reported for *E. faecium* EE3, that could be identified for up to 3 months after cessation of the probiotic [10] and for *L. fermentum* AD1-CCM7421 that persisted in lower counts for at least 5 weeks after withdrawal of oral supplementation [5]. Despite a possible prolonged survival in a few dogs, where ASV313 was detected during the post +PD washout in the current study, the duration of the 19-day washout period was probably adequate, since ASV313 could not be identified in any of the dogs fed −PD as the second diet at the end of the 16 day −PD study period. A recent review recommended the use of a minimum 4-week washout period for diet interventions with faecal microbial composition as the outcome [29], while 4 weeks would have been ideal, a longer duration might have impacted the willingness of the dog owners to adhere to study protocol. 

Recent studies suggest that breed [30,31] and neutering [32,33] might affect the faecal microbiome in dogs. The dogs included in the current study were heterogeneous in terms of breed, with only one breed (Border collie, n = 2) represented with more than one dog. At the same time the dogs acted as their own control in the cross-over design. It is therefore less likely that breed or neutering status was a major confounder in the faecal microbiota analyses in the current study. 

Several differences observed between baseline and study periods were similar irrespective of feeding the +PD or −PD diet. This indicates an effect of the dietary composition rather than the inclusion of *E. Faecium*. The dietary composition of the diets fed to the dogs prior to inclusion was not controlled. All dogs were fed commercially available diets, varying in fibre content and types which possibly resulted in differing prebiotic effects. The dietary fibres included in the current study diet mainly consisted of beet pulp, which is a fermentable fibre with a prebiotic effect that has previously been shown to affect faecal SCFA concentrations [34,35,36] and faecal microbiome composition [34,36]. The diets also contained fructooligosaccharides which has been proposed to have similar prebiotic effects as beet pulp [35]. It is therefore likely that the dietary fibre composition played a significant role in the observed effects.

In previous studies *Blautia* spp. correlated positively with increased faecal content of butyrate in dogs [37,38] and decreased inflammation in the gastrointestinal tract of dogs [38,39,40,41] and humans [42]. *Blautia* spp. might also have the ability to produce bactericins inhibiting the growth of opportunistic pathogens [43,44], such as *C. perfringens,* and in humans the genus has been attributed with the ability to promote the metabolism of unbound lipids and glucose, hereby decreasing possible obesity-associated inflammation [45]. In the current study none of the identified ASVs belonging to *Blautia* spp. correlated with faecal butyrate concentration. However, several *Blautia* spp. increased in relative abundance with *E. faecium* NCIMB 10415 incorporation into the diet, compared to the study diet without *E. faecium* NCIMB 10415 and compared to baseline. 

The *Prevotella*_9 (ASV30) was correlated with faecal butyrate concentration. *Prevotella* spp. are recognized as fibre fermenters and producers of SCFAs [46,47,48,49] and are known to increase in abundance in high fibre diets [50]. Like butyrate, faecal acetate was increased after feeding the diet, irrespective of probiotic inclusion, and the acetate concentration were observed to positively correlate with ASVs belonging to the *Romboutsia* and *Suterella* genera. The *Romboutsia* genus is a diverse genus within the Peptostreptococcaceae and at least some gut microbial strains in mice and humans are known producers of acetate, as well as formate and lactate [51,52].

Interestingly, several of the ASVs correlating with decreasing concentrations of faecal acetate are also correlating with increasing concentrations of cobalamin, namely *Fusobacterium* (ASV64), *Streptococcus* (ASV161), *Peptoclostridium* (ASV217 and ASV295), *Blautia* (ASV221), two unknown genera of the Peptostreptococcaceae family (ASV222 and ASV307), and *Peptostreptococcus* (ASV363). Previous studies have highlighted the importance of appropriate pH for *Fusobacterium*, *Streptococcus*, *Blautia*, and *Peptoclostridium* species [53,54,55,56], and a decreasing pH level has been shown to decrease their relative abundance. Furthermore, the genera *Fusobacterium*, *Streptococcus*, *Blautia*, *and Peptoclostridium* contain species with the ability to either produce or play an important role in gut microbial cobalamin production [57]. It is therefore possible, that the diet, with or without *E. faecium* NCIMB 10415, caused an increase in the relative abundance of acetate producing microbes, which have decreased the relative abundance of pH sensitive producers of cobalamin.. This could explain how, although within reference concentrations, serum cobalamin concentrations were significantly decreased after both feeding periods. It is therefore possible that the observed cobalamin lowering effect is related to changes in the dietary prebiotic/fibre content instead of the *E. faecium* NCIMB 10415 incorporation. However, due to a detectable prolonged survival during the washout period of the dogs fed the +PD diet first, it is not possible to completely exclude a possible cobalamin lowering effect of *E. faecium* NCIMB 10415 in the current study. In a study investigating the effect of supplementing a symbiotic containing *E. faecium* SF68, psyllium and brewer’s yeast to healthy adult dogs, circulating cobalamin and folate, 8/18 dogs were classified with hypocobalaminenia 14 days after withdrawal of the supplement [13]. The authors proposed that the microbiome consumption of cobalamin is increased either by *E. faecium* itself or other microbiome bacteria, for which the abundance has been positively affected by *E. faecium* symbiotic supplementation [13]. Since the cobalamin lowering effect observed in the current study was associated with diet irrespective of probiotic inclusion or order of study periods, it is possible that the change in diet alone may disrupt the balance of the gut microbiome enough to inadvertently decrease relative abundance of cobalamin producing microbes. 

In contrast to the effect on serum cobalamin, only feeding with the diet containing *E. faecium* NCIMB 10415 had a serum cholesterol lowering effect. Similar to the current study, supplementation with *Lactobacillus johnsonii* CPN23 (0.1 mL/kg BW, 108 CFU/mL) or *Lactobacullus acidophilus* NCDC15 (0.1 mL/kg BW, 108 CFU/mL) for 9 weeks, resulted in lower circulating cholesterol and triglyceride concentrations compared with controls [11]. Additionally, a study investigating the effect of *E. faecium* SF68 on hepatic biomarkers in 18 healthy adult dogs, observed a lowering effect on circulating cholesterol that remained within the reference range 14 days after cessation of supplementation [58]. Another strain of *E. faecium* (*E. faecium* EE3) has been reported to normalise both high and low serum cholesterol in clinically unaffected dogs after supplementation for 1 week [10]. In contrast, supplementation with *Lactobacillus fermentum* AD1 (3 × 10^9^ CFU/mL) and *Bifidobacterium animalis* B/12 (1.04 × 10^9^ CFU/mL) did not affect circulating cholesterol concentrations of healthy dogs [8,12]. These findings illustrate an inherent difference in specific effect between probiotic species and strains, with formulation and dosage being another possible contributing factor. 

Some microbes are known to enhance bile acid conjugation within the intestinal tract, which could explain the serum cholesterol lowering effect of certain probiotics. This leads to impaired reabsorption of bile acid into the entero-hepatic circulation and thus increased de novo host production of bile acid for which cholesterol is a precursor [59,60]. Incorporating or binding of cholesterol to the microbiome cell membrane could also play a role [61]. In hamsters fed a high cholesterol diet, the presence of butyrate, propionate and acetate in the gastrointestinal tract decreased circulating cholesterol concentration by enhancing faecal bile acid secretion and upregulating bile acid synthesis [62]. Studies in different mammalian species further suggest that propionate [63,64] and acetate [65] are capable of decreasing cholesterol production through hepatic enzymatic inhibition. In the present study, the increased faecal concentration of acetate and butyrate in +PD fed dogs compared to baseline could suggest that increased acetate and butyrate might also play a role in the cholesterol lowering effect. However, faecal acetate and butyrate was also significantly increased in the −PD fed dogs without significantly decreasing cholesterol concentrations compared to baseline. 

## 5. Conclusions

This study investigated metabolic and faecal microbial effects of a commercially available diet containing the probiotic *E. faecium* NCIMB 10415 10^9^ CFU/kg for adult dogs by comparing baseline at study start and nutritionally similar diets manufactured with or without incorporation of the probiotic. Results suggest that in some dogs, *E. faecium* NCIMB 10415 can possibly survive in the canine gut microbiome for 19 days after withdrawal of oral supplementation. Analyses of the microbial diversity and changes in relative abundance of some specific microbes indicate that *E. faecium* NCIMB 10415 has a mild positive effect on the microbiome and owners also reported mildly improved faecal quality when comparing the diet with or without *E. faecium* NCIMB 10415. In addition, the probiotic exerted a mild cholesterol lowering effect, which might be favourable in conditions associated with hypercholesterolemia. 

Despite a small possibility of prolonged survival of *E. faecium* NCIMB 10415 in the gut, beyond the 19 days of washout, the observed increase of faecal butyrate and acetate, and decreased circulating cobalamin, in both feeding periods, are most likely attributable to properties of the study diet alone. The effect on cobalamin was seen with both the diet with and without *E. faecium* NCIMB 10415, and the effect was mild and no hypocobalaminemia developed as described in previous probiotic studies. 

## Figures and Tables

**Figure 1 animals-13-00144-f001:**
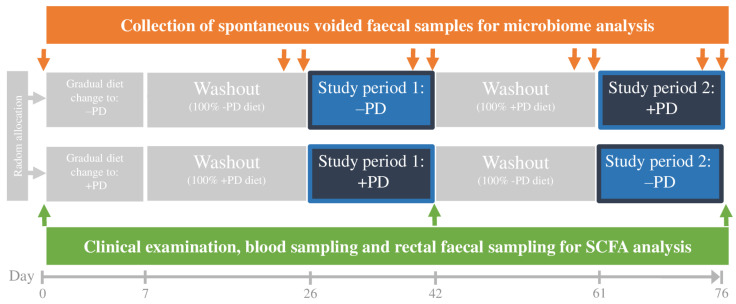
Study design. The study was a prospective randomized double-blinded crossover study composed of two study periods of each 16 days, both with a preceding washout period of 19 days. Included dogs (n = 11) were exclusively fed the same nutritionally balanced commercial canine diet with (+PD) or without (−PD) *Enterococcus faecium* NCIMB 10415 10^9^ CFU/kg incorporated. Orange arrows indicate time points for collection of spontaneously voided faecal sampling for microbiome analysis (day 0, 24, 25, 40, 41, 59, 60, 75 and 76). Green arrows indicate time points for collection of blood samples for serum triglyceride, cholesterol, cobalamin, and folate concentrations as well as rectal faecal sampling for short chain fatty acids (SCFA) analysis (day 0, 42 and 76).

**Figure 2 animals-13-00144-f002:**
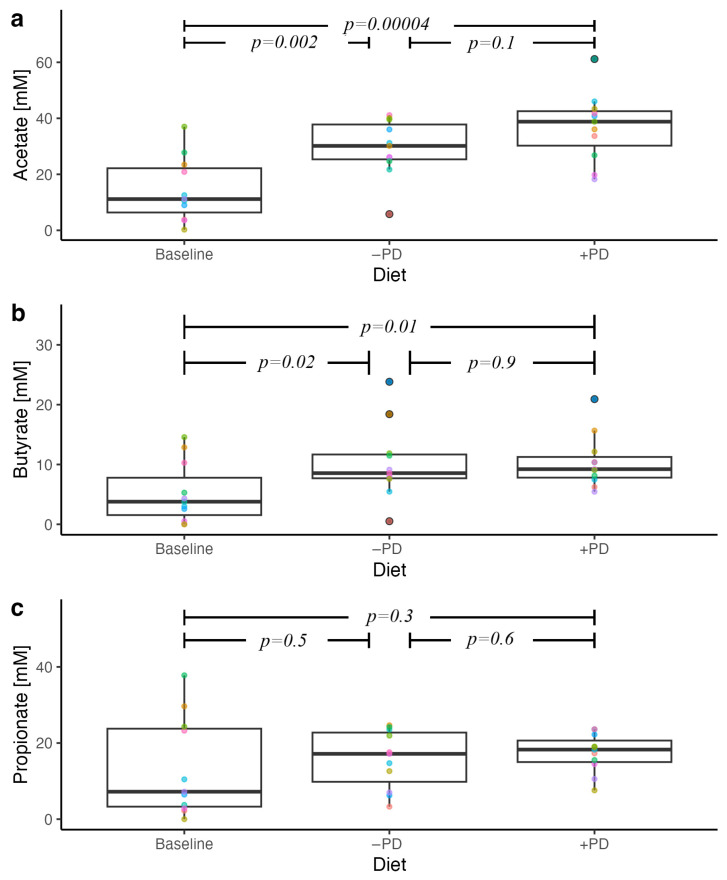
Faecal short chain fatty acid concentrations in privately owned healthy adult dogs (n = 11) fed a commercial diet with (+PD) or without (−PD) *Enterococcus faecium* NCIMB 10415 10^9^ CFU/kg for 16 days each in a crossover study design compared to baseline concentrations. Each dot colour on the plot identifies the individual dog across diets. *p* values < 0.05 were considered significant.

**Figure 3 animals-13-00144-f003:**
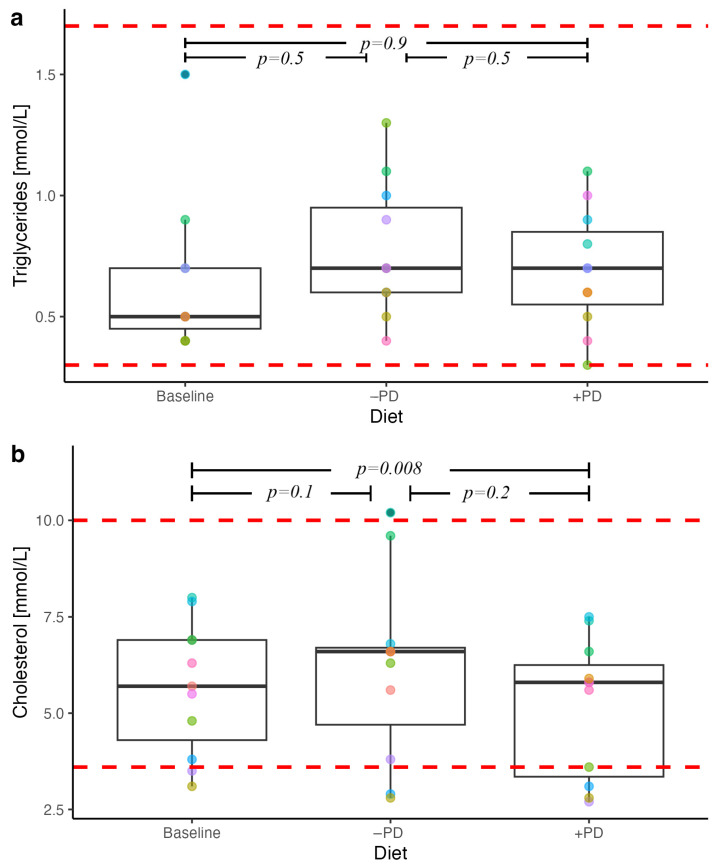
Serum lipid concentrations in privately owned healthy adult dogs (n = 11) fed a commercial diet with (+PD) or without (−PD) *Enterococcus faecium* NCIMB 10415 10^9^ CFU/kg for 16 days each in a crossover study design compared to baseline concentrations. Each dot colour on the plot identifies the individual dog across diets. Red dotted lines represent laboratory reference ranges. *p* values < 0.05 were considered significant.

**Figure 4 animals-13-00144-f004:**
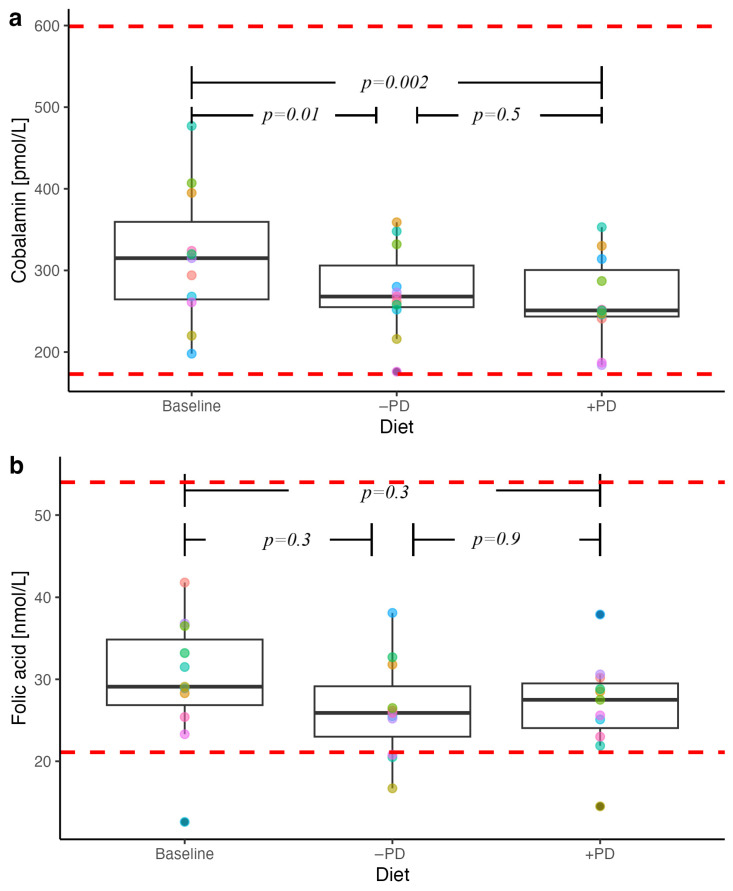
Serum B-vitamin concentrations in privately owned healthy adult dogs (n = 11) fed a commercial diet with (+PD) or without (−PD) *Enterococcus faecium* NCIMB 10415 10^9^ CFU/kg for 16 days each in a crossover study design compared to baseline concentrations at inclusion. Each dot colour on the plot identifies the individual dog across diets. Red dotted lines represent laboratory reference ranges. *p* values < 0.05 were considered significant.

**Figure 5 animals-13-00144-f005:**
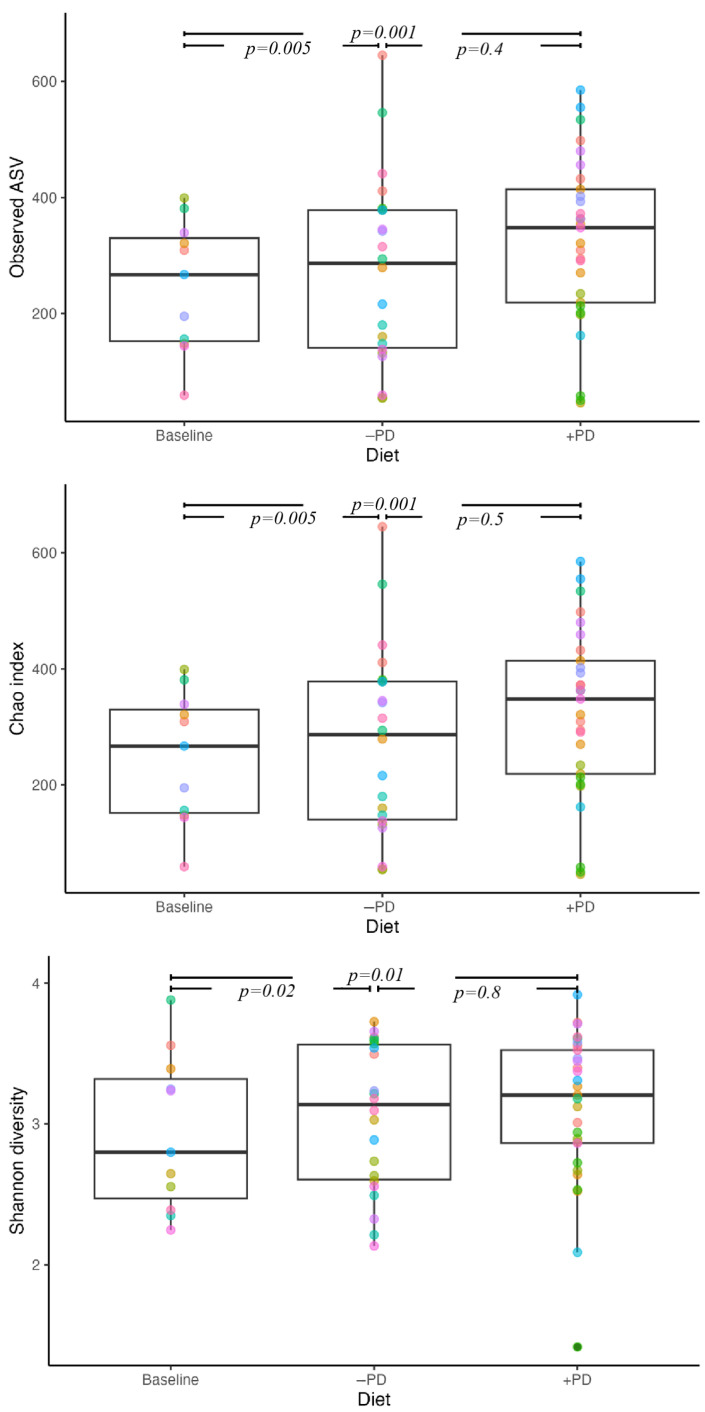
Faecal microbiome alpha diversity in privately owned healthy adult dogs (n = 11) fed a commercial diet with (+PD) or without (−PD) *Enterococcus faecium* NCIMB 10415 10^9^ CFU/kg for 16 days each, in a crossover study design. Boxplots showing the observed richness (Observed ASV), Chao1 and Shannon index, of the faecal microbiome when the study population was fed a diet manufactured with (+PD) or without (−PD) *Enterococcus faecium* NCIMB 10415 as compared to baseline diversity at inclusion. Each number on the plot identifies the individual dog across diets.

**Figure 6 animals-13-00144-f006:**
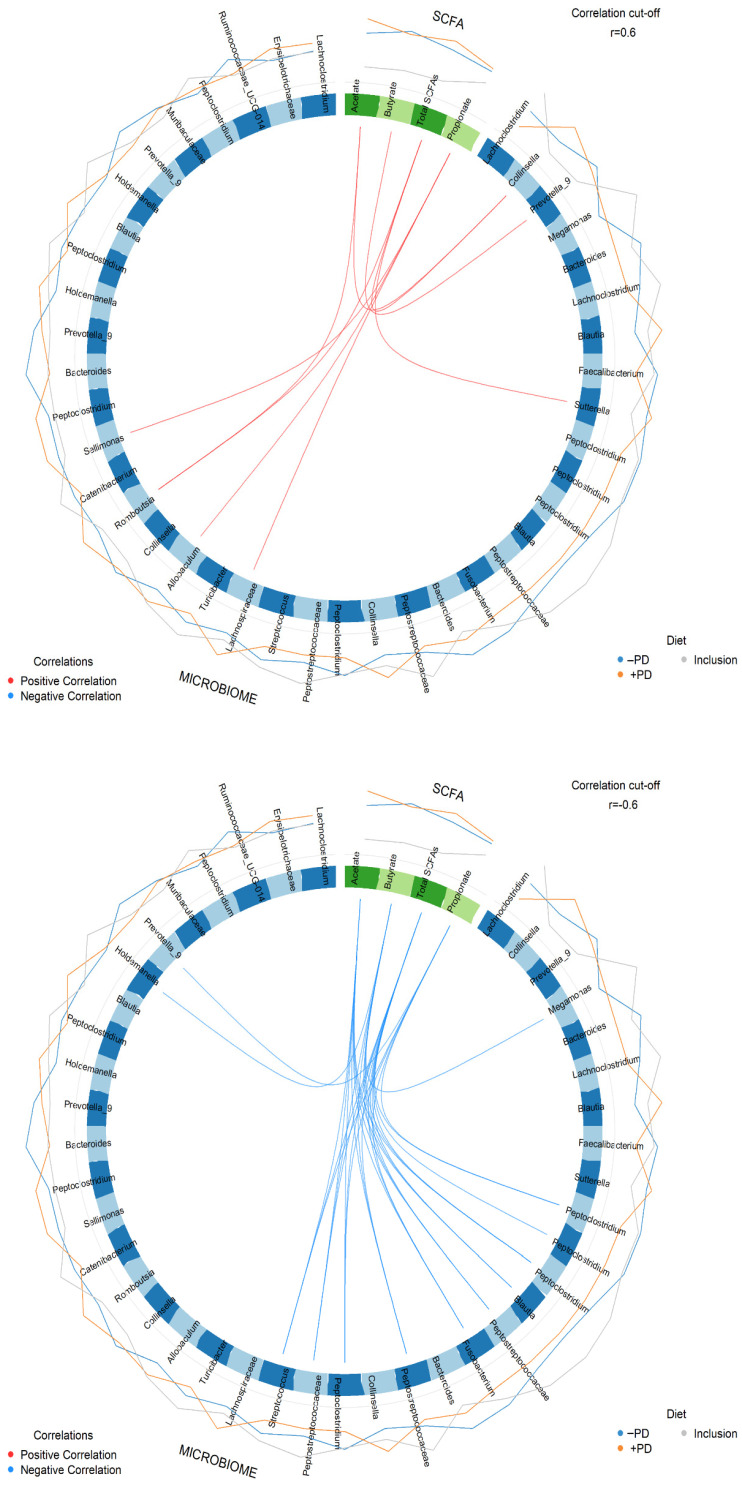
Correlation between microbial abundance and faecal acetate, propionate, butyrate and total short chain fatty acids concentration (SCFA). Circos plot illustrating positive correlations (r ≥ 0.6, connecting red lines) and negative correlations (r ≤ −0.6, connecting blue lines) performed using multiblock sparse partial least squares discriminant analysis of faecal acetate, propionate, butyrate and total SCFAs, and relative abundance of faecal microbes at study inclusion (baseline samples, circumferential grey line), feeding a diet without (−PD, circumferential blue line) or with 10^9^/kg *Enterococcus faecium* NCIMB 10415 (+PD, circumferential orange line).

**Figure 7 animals-13-00144-f007:**
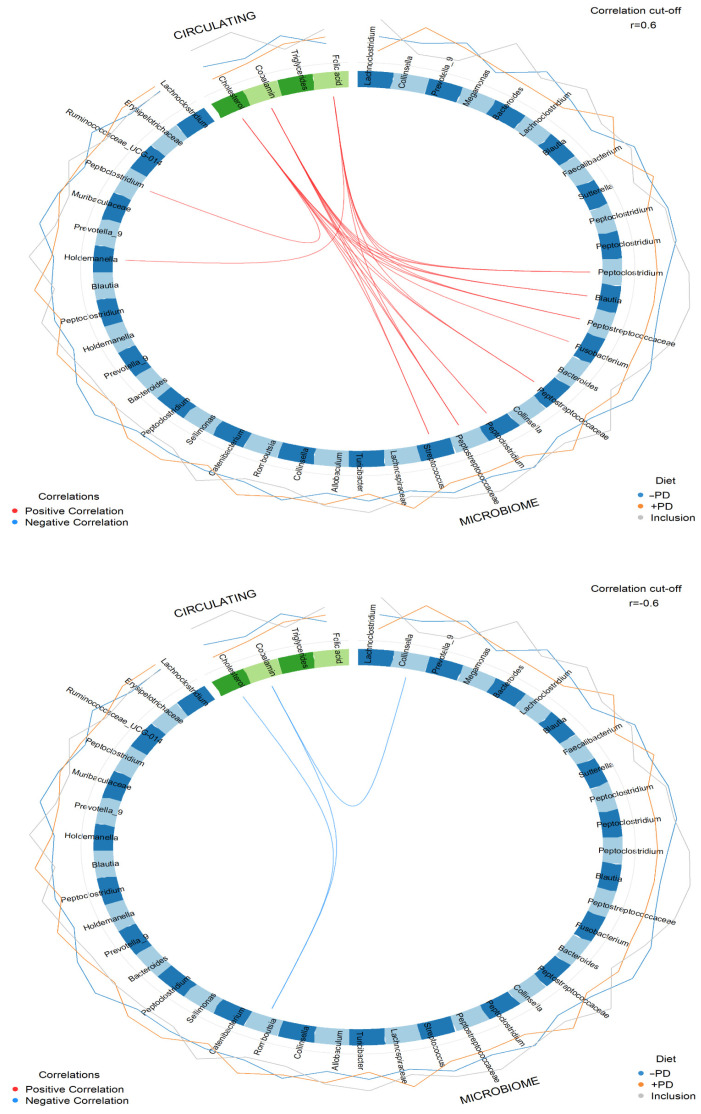
Correlation between faecal microbial abundance and serum cobalamin, folate, cholesterol, and triglycerides. Circos plot illustrating positive correlations (r ≥ 0.6, connecting red lines) and negative correlations (r ≤ −0.6, connecting blue lines) performed using multiblock sparse partial least squares discriminant analysis of the serum concentrations of cobalamin, folate, cholesterol and triglycerides, and relative abundance of faecal microbes at study inclusion (baseline concentrations, circumferential grey line), when feeding a diet without (−PD, circumferential blue line) or with 10^9^/kg *Enterococcus faecium* NCIMB 10415 (+PD, circumferential orange line).

**Table 1 animals-13-00144-t001:** Nutritional composition of study diets.

Content	− PD	+ PD
Metabolisable energy ^a^	17,364 kJ/kg (4150 kcal/kg)	17,364 kJ/kg (4150 kcal/kg)
Crude protein ^a^	25%	25%
Crude fat ^a^	15%	15%
Carbohydrate ^a^	44%	44%
Crude ash ^a^	7%	7%
Crude fibre ^a^	2.2%	2.2%
*E. faecium* NCIMB 10415 ^b^	<10 CFU/kg ^c^	3.8–5.0 × 10^9^ CFU/kg

Dietary composition of two dry kibble diets used in the present study. −PD; control diet without added probiotic, +PD; probiotic diet, the same diet as the control diet with 3.8–5.0 × 10^9^ CFU/kg *Enteroccocus faecium* NCIMB 10415 incorporated. CFU: cell forming units. ^a^ As stated by the manufacturer of the diet (Bacterfield GmBH, Hamburg, Germany). ^b^ Analysis by ALS Laboratories, UK prior to the study. ^c^ Below detectable concentration.

**Table 2 animals-13-00144-t002:** Effect of diets on relative bacterial abundance in faecal microbiota.

Genus	ASV	+PD Compared to Baseline			−PD Compared to Baseline			+PD Compared to −PD		
		Estimated Effect	Std. Error	FDR	Estimated Effect	Std. Error	FDR	Estimated Effect	Std. Error	FDR
*Lachnoclostridium*	ASV32	−3.5	0.67	2.8 × 10^−4^	−3.2	0.78	0.007			
*Prevotella_9*	ASV82	−3.3	0.61	2.8 × 10^−4^	−2.8	0.64	0.006			
*Blautia*	ASV41	4.2	1.1	0.019						
*Enterococcus*	ASV313	2.9	0.81	0.031				2.5	0.45	0.03
*Enterococcus*	ASV335	−6.2	1.8	0.043	−8.4	2.3	0.017			
*Bacteroides*	ASV110				3.0	0.64	3.4 × 10^−4^	−1.0	0.26	0.0098
Lachnospiraceae_UCG−010	ASV332				4.3	1.1	0.007			
*Bacteroides*	ASV153				4.7	1.3	0.023	−2.5	0.50	8.5 × 10^−4^
*Bacteroides*	ASV63				2.0	0.58	0.028	−1.4	0.25	1.3 × 10^−4^
*Bacteroides*	ASV102				3.0	0.90	0.028			
*Bacteroides*	ASV149							−3.7	0.83	0.0030
*Ruminiclostridium_9*	ASV436							−2.7	0.62	0.0032
*Holdemania*	ASV356							−1.9	0.48	0.0067
*Blautia*	ASV52							0.92	0.24	0.0079
*Blautia*	ASV24							0.71	0.20	0.0098
*Bacteroides*	ASV75							−1.8	0.50	0.0098
Unknown Eggerthellaceae	ASV84							−1.3	0.34	0.0098
*Blautia*	ASV101							1.5	0.43	0.0098
*Lachnospira*	ASV16							−1.5	0.41	0.010
*Lachnoclostridium*	ASV116							−1.0	0.33	0.027

The effect of a commercial diet with (+PD) or without (−PD) *Enterococcus faecium* NCIMB 10415 10^9^ CFU/kg, for 16 days each on, relative bacterial abundance in faecal microbiota in client owned healthy dogs (n = 11). False discovery rate (FDR) < 0.05 was considered significant. ASV; Amplicon Sequence Variant.

**Table 3 animals-13-00144-t003:** Faecal *Enterococcus* spp.

Order of Diet	-	+PD Followed by −PD				−PD Followed by +PD			
Time of Sampling	Baseline	Washout Baseline	+PD Period	Washout +PD	−PD Period	Washout Baseline	−PD Period	Washout −PD	+PD Period
*Enterococcus* ASV313	0/11	3/23	9/23	3/22	0/22	0/19	0/17	0/18	9/19
*Enterococcus* ASV335	1/11	0/23	0/23	0/22	1/22	0/19	0/17	0/18	0/19
*Enterococcus* ASV796	1/11	2/23	2/23	0/22	0/22	0/19	0/17	0/18	0/19

Number of dogs that had the three *Enterococcus* genera ASV313, ASV335, and ASV796 present in their faecal microbiome at various point of time during the study and depending on whether they were fed a diet without probiotic (−PD) prior to or after being fed at diet containing 10^9^/kg *E. faecium* NCIMB 10415 (+PD).

## Data Availability

The datasets used and/or analyzed during the current study are available at either UCPH ERDA data (https://sid.erda.dk/sharelink/eZ0NFSju9q) or at the EMBL Nucleotide Sequence Database (ENA), with the study accession number PRJEB55896.

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
