# Peer review of "Alterations in Healthy Adult Canine Faecal Microbiome and Selected Metabolites as a Result of Feeding a Commercial Complete Synbiotic Diet with Enterococcus faecium NCIMB 10415"

_animals, 2022, doi:10.3390/ani13010144_

Round 1
Reviewer 1 Report
Summary: Fascinating article which discusses an impactful topic in companion animal nutrition, probiotics. Studies like this are complicated due to the lack of control on the test subjects and to be able to get a number of clients to diligently work on this project for 3 months or so is very commendable. It is very interesting to see the changes in microbiome, SCFA’s, and fecal consistency observed and I definitely believe this is a huge area of research interest, as well as public interest as many individuals in the world enjoy having companion animals such as dogs.
General comments about the article: Given some of the obvious challenges with this type of research, I appreciate all of your efforts to conduct the project as well as identifying pet owners to contribute to the data. I am curious why no samples of any kind were taken during the first washout period, after the initial 7-day adaption period? There seems to have been a tremendous opportunity lost by having 40 days from inclusion before any further samples were collected (fecal or blood). Though the washout periods are good, do you have any thoughts on if / how much breed or sex of the study dogs may have played into your results, especially the microbiome work? Was any separate analysis done to confirm that the Enterococcus ASV313 was in fact the E. faecium NCIMB 10415 which was given in the +PD diets? If not, though I agree that it seems highly likely it is, I would caution you about concluding that “it persists” for 19 days following cessation of administration.
General comments about the review: Generally, I think the review was thorough and covered the previous research well. I do believe a discussion or at least acknowledgement, that breed may impact the microbiome is necessary and I might suggest looking at the paper by Reddy et al., 2019 (https://doi.org/10.4014/jmb.1906.06048) as well as the You and Kim, 2022 (Reference # 38) paper for this topic. Also, there is a paper by Jarett et al., 2021 (https://doi.org/10.3389/fvets.2021.644836) which may be useful in confirming some of the elements of your study design for the microbiome work.
Specific comments: Italicize all p-values (p=) in the manuscript
- On Line 88, change “has” to “have”
- On Line 177, should be “fructooligosaccharides”
- On Line 179, add an “a” in technic in
- On Line 282, consider changing p=4×10-5 to p<0.001
- Double check all references are cited and correct
Reviewer 2 Report
The INTRODUCTION is too large. Please sammary the information from line 63 to line 86. Too detailed informations about Lactobacillus and this bacteria wasn't the topic of the manuscript. The studies that you mentioned in the INTRODUCTION (i.e. references no. 7, 8, 11, 12, 13, etc...) should be moved to DISCUSSIONS.
Also, the MATERIALS AND METHODS are too large. Please make a summary. Are hard to read and to follow the experiment. Please try to include some subchapters here for MATERIALS AND METHODS. For example 2.1. Study design, 2.2. Animals used, and so one..... as you did for RESULTS.
The nutritional composition of the studied diet you copied from the label or you did the chemical analysis of the diet? It is important to mention.
In my oppinion also the DISCUSSION are too large.
This is a hard to read manuscript. It has to be more summarised. The information should be better structured in subchapters to be mare easy to read it.
The manuscript contain valuable informations that should be useful for vets, owners and other researchers in the fied. Taking into account that only eleven dogs were used for this study, I think that future studies are needed on this topic.
Round 2
Reviewer 1 Report
I appreciate your revised manuscript and believe that it more clearly reflects the scope, limitations, and important results of your study. I have no major issues with this revised version.
On Line 130, I would change "was" to "were"